# Beyond Seizure Control: Treating Comorbidities in Epilepsy via Targeting of the P2X7 Receptor

**DOI:** 10.3390/ijms23042380

**Published:** 2022-02-21

**Authors:** Beatriz Gil, Jonathon Smith, Yong Tang, Peter Illes, Tobias Engel

**Affiliations:** 1Department of Physiology and Medical Physics, Royal College of Surgeons in Ireland, University of Medicine and Health Sciences, D02 YN77 Dublin, Ireland; beatrizgil@rcsi.ie (B.G.); jonathonsmith@rcsi.ie (J.S.); 2FutureNeuro, Science Foundation Ireland Research Centre for Chronic and Rare Neurological Diseases, Royal College of Surgeons in Ireland, University of Medicine and Health Sciences, D02 YN77 Dublin, Ireland; 3International Collaborative Centre on Big Science Plan for Purinergic Signaling, Chengdu University of TCM, Chengdu 610075, China; tangyong@cdutcm.edu.cn (Y.T.); peter.illes@medizin.uni-leipzig.de (P.I.); 4School of Acupuncture and Tuina, Chengdu University of TCM, Chengdu 610075, China; 5Rudolf Boehm Institute for Pharmacology and Toxicology, University of Leipzig, 04107 Leipzig, Germany

**Keywords:** epilepsy, comorbidities, purinergic signalling, ATP, P2X7 receptor

## Abstract

Epilepsy is one of the most common chronic diseases of the central nervous system (CNS). Treatment of epilepsy remains, however, a clinical challenge with over 30% of patients not responding to current pharmacological interventions. Complicating management of treatment, epilepsy comes with multiple comorbidities, thereby further reducing the quality of life of patients. Increasing evidence suggests purinergic signalling via extracellularly released ATP as shared pathological mechanisms across numerous brain diseases. Once released, ATP activates specific purinergic receptors, including the ionotropic P2X7 receptor (P2X7R). Among brain diseases, the P2X7R has attracted particular attention as a therapeutic target. The P2X7R is an important driver of inflammation, and its activation requires high levels of extracellular ATP to be reached under pathological conditions. Suggesting the therapeutic potential of drugs targeting the P2X7R for epilepsy, P2X7R expression increases following status epilepticus and during epilepsy, and P2X7R antagonism modulates seizure severity and epilepsy development. P2X7R antagonism has, however, also been shown to be effective in treating conditions most commonly associated with epilepsy such as psychiatric disorders and cognitive deficits, which suggests that P2X7R antagonisms may provide benefits beyond seizure control. This review summarizes the evidence suggesting drugs targeting the P2X7R as a novel treatment strategy for epilepsy with a particular focus of its potential impact on epilepsy-associated comorbidities.

## 1. Epilepsy: Current Treatments and Shortcomings

The term epilepsy comprises a heterogeneous group of brain disorders that all share in common an enduring predisposition to generate spontaneous, recurrent seizures. Epilepsy is one of the most common chronic diseases of the central nervous system (CNS), with an incidence of 1–2% within the general population. This amounts to over 70 million people diagnosed with epilepsy worldwide, disproportionally affecting the young and the elderly [1,2,3]. Further reducing the quality of life, the risk of premature death is 2–3 times higher in patients with epilepsy than the general population and, as discussed in detail within the next sections, a significantly increased risk of additional comorbidities, such as anxiety or depression [2,4,5]. While in the majority of cases the exact causes of epilepsy remain unknown, epilepsy can result from genetic abnormalities (e.g., Dravet syndrome) or can be acquired following a precipitating injury. Genetic causes include polymorphisms, deletions, and duplications that affect DNA copy numbers, or de novo mutations in genes [6,7]. Acquired epilepsies involve a precipitating insult to the brain such as status epilepticus, traumatic brain injury (TBI), stroke, glioblastoma, infection to the brain, and metabolic disturbances [8,9,10,11,12]. Epileptogenesis, triggered by an injury to the brain, refers to the process of transforming a normal brain into a brain experiencing epileptic seizures [13]. Pathological processes occurring during epileptogenesis include increased permeability of the blood–brain barrier (BBB), changes in structural and synaptic plasticity, neurodegeneration, aberrant neurogenesis, epigenetic changes, and a sustained activation of inflammatory processes [14,15,16].

Treatment for epilepsy includes pharmacological interventions via anti-seizure drugs (ASDs) and alternative approaches such as surgical resection of the epileptic focus, use of the ketogenic diet, and vagus nerve or deep brain stimulation [1]. To date, ASDs remain the first-line treatments for epilepsy. The mechanism of action of these drugs is mainly to promote inhibitory γ-aminobutyric acid (GABA)ergic neurotransmission (e.g., lorazepam, midazolam, phenobarbital), to inhibit excitatory glutamatergic neurotransmission (e.g., topiramate, felbamate), or to block voltage-gated sodium channels resulting in reduction of neurotransmission (e.g., phenytoin, lamotrigine) [17]. However, while ASDs can manage seizures in epilepsy in the majority of cases, approximately 30% of patients remain drug-refractory [1,17]. Temporal lobe epilepsy (TLE), which can be acquired following an injury to the brain and includes structural, physiological, biochemical, and epigenetic alterations within the limbic system, is the most common form of epilepsy in adults and is particularly prone to drug-refractoriness [16,18]. Of note, TLE is also one of the most studied forms of epilepsy in the setting of purinergic signalling [19]. In addition to drug refractoriness, ASDs have a high burden of adverse effects (e.g., fatigue, irritability, dizziness), which are thought to be merely symptomatic without noticeable impacts on disease progression. Consequently, there is a persistent need to develop new ASDs that act upon non-classical mechanisms of action, effective in drug-refractory patients, with a disease-modifying potential and the capacity to spare patients from unwanted side effects.

Mounting evidence demonstrates that purinergic signalling via extracellularly released adenosine triphosphate (ATP) plays an important role during several shared pathological conditions among brain diseases (e.g., inflammation, hyperexcitability, and cell death) and, therefore, constitutes a possible common pathological mechanism underlying primary diseases of the brain and their associated comorbidities [20,21]. Once released into the extracellular space, ATP activates specific purinergic receptors, including the cationic P2X7 receptor (P2X7R), which has attracted the most attention among the ATP-gated receptors as a drug target for the treatment of diseases of the CNS [22,23,24]. P2X7R antagonism has not only been shown to suppress seizures and epilepsy [25,26] but has also shown beneficial effects in several other neurological conditions commonly associated with epilepsy, such as depression, psychosis, or dementia, among many others [27,28,29]. In the following sections, we will provide an up-to-date summary of the evidence suggesting a contribution of the ATP-P2X7R axis to seizures and epilepsy with a particular focus on epilepsy-associated comorbidities.

## 2. Comorbidities in Epilepsy

According to Feinstein et al. [30], comorbidities are defined as “any distinct additional entity that has existed or may occur during the clinical course of a patient who has the index disease (i.e., epilepsy) under study.” When referring to epilepsy, it is important to keep in mind that epilepsy, rather than constituting a uniform entity, represents a spectrum of different brain conditions, with differences in aetiology, clinical manifestations, demographics, treatment responses, and prognosis [31]. It therefore comes as no surprise that, in addition to epileptic seizures, patients with epilepsy suffer from a wide range of additional conditions including psychiatric illnesses, cognitive impairment, sleep disorders, cardiovascular disease, bone diseases, or premature mortality, among many others (Figure 1). Some conditions are up to eight times more common in patients with epilepsy compared to the general population, with up to ~80% of patients presenting at least one comorbid medical condition [5,32]. These comorbidities represent a substantial burden for people with epilepsy as comorbid conditions often have a greater effect on quality of life than do seizures. Comorbidities can arise due to seizures themselves, adverse effects of ASDs or other treatments, or shared underlying pathologies and aetiologies, with the latter suggesting that drugs acting on shared pathological processes (e.g., underlying inflammation) may provide the opportunity to target both primary disease and associated comorbidities. Below, we will provide a summary of the most common comorbidities in epilepsy with a particular focus on comorbidities associated with the brain. For more detailed information, please refer to [32,33,34].

Psychiatric disorders are one of the most frequently reported comorbidities, estimated to occur in 25–50% of patients with epilepsy [4], and are particularly prevalent in patients with TLE and drug-refractory epilepsy [35,36,37,38]. Several factors have been described to contribute to the presence of psychiatric disorders in patients with epilepsy, including treatment with ASDs and shared underlying pathologies. Among psychiatric disorders, the most frequent ones are depression, anxiety, and psychotic disturbances. Almost 40% of patients with epilepsy show some form of depression, making this the most common comorbidity [39]. Conversely, patients with epilepsy and concurrent depression present a higher risk of developing drug-resistant epilepsy than those without depression [40,41,42], and several studies have suggested that depression can increase the risk of developing epilepsy [43,44]. Moreover, suggesting shared underlying pathologies, both conditions, epilepsy and depression, share a considerable number of features, including their episodic nature, the efficacy of ASDs for the treatment of both disorders, and the similarities in network activity found by neuroimaging evidence [45,46,47]. It is interesting to note that the antiepileptics phenytoin and lamotrigine are used as therapeutics for major depression or the depressive phase of bipolar disorder. Following depression, anxiety has been described as the second most common comorbidity in epilepsy patients and is especially prevalent in patients suffering from drug-resistant TLE [48,49]. Anxiety can occur not only as a reaction to the diagnosis of epilepsy but also as a consequence of epilepsy or as a side effect of ASDs. Demonstrating its detrimental effects, epilepsy patients with anxiety reported more severe epilepsy and an overall lower quality of life [49]. Although less frequent, psychotic disorders are another serious comorbidity in epilepsy (~5% in epilepsy patients and up to 7% in patients with TLE) [50]. Studies have demonstrated a complex bidirectional association between epilepsy and psychotic disorders. It is known, for example, that patients with schizophrenia have a 2–3-fold increased risk of developing epilepsy [51,52]. In other cases, psychosis can be prevented with seizure control, while in others, certain ASDs can worsen the psychotic condition, and some antipsychotic treatments can aggravate epileptic seizures [53,54]. In addition, patients with one condition are at greater risk of developing the other [55]. Dementia is another commonly associated comorbidity in patients with epilepsy with recurrent seizures, leading to the impairment of cognition and progressive memory loss [56,57]. Of note, patients with dementia, such as Alzheimer’s disease (AD), are at an increased risk of developing epilepsy, with both diseases, epilepsy and AD, sharing similar underlying pathologies [58,59,60]. How AD pathology leads to epileptic seizures has not been fully established; however, a major role for both the amyloid-β peptide and the microtubule-associated tau protein has been suggested [61,62].

Other commonly diagnosed comorbidities in epilepsy include migraine, cardiovascular diseases, stroke, respiratory diseases (e.g., asthma), gastrointestinal conditions, allergies, bone diseases, and increased premature mortality [33,63,64,65,66,67,68,69,70,71]. Diabetes represents another comorbidity, with epilepsy patients being at a higher risk of contracting diabetes and adverse postdiabetes outcomes [72].

Demonstrating the importance of treating both primary conditions (i.e., epilepsy) and associated comorbidities, comorbidities may be linked to poor seizure control (e.g., migraine, psychiatric disorder) and may be reasons for a further reduction in the quality of life (e.g., depression) [73,74]. Notably, increased brain excitability and seizures are frequently associated with other brain diseases, including neurodegenerative ones (e.g., AD [75,76] and Huntington’s disease [77]) and psychiatric diseases (e.g., schizophrenia [51]), where they may contribute to disease progression [78,79]. Consequently, treatment with ASDs suppressing a pathological brain hyperexcitability may be beneficial not only in the context of epilepsy, but also for other non-epileptic CNS conditions (e.g., psychiatric disorders) [80].

## 3. Purinergic Signalling—Overview

The purinergic signalling system represents probably the most primitive and widespread chemical messenger system in animals [81] and comprises a myriad of enzymes, transporters, and receptors, which will be briefly described below (Figure 2). For a more detailed description, please refer to [21,82]. It is now well established that the nucleotide ATP, mainly known as the main cellular energy currency, acts also as a signalling molecule in the brain [83]. Usually found at very low extracellular concentrations, ATP levels rapidly increase under pathological conditions (e.g., inflammation, increased neuronal activity, and cell death) acting as damage-associated molecular patterns (DAMPs) [84]. ATP can be released from neurons and glia in an uncontrolled fashion (e.g., from damaged cells) or via exocytosis or non-exocytotic mechanisms including transporters (e.g., ATP-binding cassette (ABC) transporters), membrane channels such as P2X7Rs, and furthermore pannexin-1 channels and connexin-43 hemichannels [85]. Following its release, ATP is rapidly metabolized by ectonucleotidases including ectonucleoside triphosphate diphosphohydrolase family (E-NTPDases), ectonucleotide pyrophosphatase/phosphodiesterase family (E-NNP), and alkaline phosphatases (ALP) into different breakdown products (e.g., adenosine diphosphate (ADP), adenosine) [86,87]. ATP accumulates at injured sites, activating purinergic receptors, including P2X7Rs [88]. ATP releases into the extracellular space is now widely acknowledged as a shared mechanism during the pathogenesis of a wide array of brain diseases [20].

Purinergic receptors are divided into the adenosine-sensitive P1 receptors and P2 receptors, which are activated by extracellular adenine and uridine nucleotides (i.e., ATP or uridine triphosphate (UTP)). For a more detailed description of P1 receptors, which, to date, include four receptor subtypes (A_1_, A_2A_, A_2B_, and A_3_), and their role in epilepsy, please refer to [89,90]. P2 receptors are divided into metabotropic P2Y receptors (P2YR) and ionotropic P2X receptors (P2XR). To date, we recognize eight different P2YRs that have the typical seven transmembrane segment of G-protein-coupled receptors, which are mainly activated by ATP (P2Y2, P2Y11), adenosine diphosphate (ADP) (P2Y1, P2Y12, and P2Y13), UTP/uridine diphosphate (UDP) (P2Y2, P2Y4, and P2Y6), and the ligand UDP-glucose (P2Y14) [91,92]. P2XRs, activated by ATP, form a ligand-gated ion channel, allowing the passage of the positively charged ions Na^+^, Ca^2+^, and K^+^. The P2XR family consists of seven members, which form a trimeric structure that can be either homomeric or heteromeric according to the recruitment of identical or diverse receptor-subtypes [93].

As mentioned before, among the purinergic receptors, the P2X7R has attracted most attention as a drug target for brain diseases with mounting evidence showing its causal involvement in multiple chronic diseases of the CNS [22,24,94]. Notably, the P2X7R has a low affinity for ATP compared with other purinergic receptors, suggesting that its activation occurs mainly under pathological conditions of high ATP release. P2X7R-based treatment is therefore presumed to cause less adverse side effects, preventing actions on normal networks. In addition, P2X7Rs have slow desensitization dynamics and the ability to permeabilize the cell membrane to molecules up to 900 Daltons in size [95,96]. The P2X7R has been described as a gatekeeper of inflammation. P2X7R activation is an essential step in the activation and proliferation of microglia and a key regulator of the nucleotide-binding oligomerization domain-, LRR- and pyrin domain-containing protein 3 (NLRP3) inflammasome, inducing the release of proinflammatory cytokines (e.g., interleukin-1β (IL-1β) or IL-18) [88,97]. In addition, the human P2X7R is a highly polymorphic gene with several single-nucleotide polymorphisms (SNPs) shown to change receptor function, either as loss- or gain-of-function variants [98]. Importantly, several of these P2X7R polymorphisms have been associated with common comorbidities associated with epilepsy, as discussed in more detail below. P2X7R activation has, however, also been linked to other damaging processes occurring during brain diseases, such as the modulation of neurotransmitter release (e.g., glutamate), cell death, BBB opening, increased hyperexcitability, and synaptic plasticity [95,96]. The presence of P2X7Rs has been confirmed throughout the brain. While there is broad agreement regarding its expression on microglia and oligodendrocytes, whether P2X7Rs are expressed and functional on neurons and astrocytes remains a matter of debate [99,100,101,102]. Nevertheless, regardless of its cell-type-specific expression, treatments based on P2X7R antagonisms have provided promising results for several brain conditions, including the most prominent epilepsy-associated comorbidities as discussed within the next section.

## 4. Targeting of the P2X7R and Seizure Control

The heterologous nature of epilepsy is reflected in the abundance of experimental models available to mimic both acute seizures (e.g., status epilepticus) and epilepsy. This includes the use of seizure-inducing chemoconvulsants injected systemically or directly into the brain (e.g., kainic acid (KA), pilocarpine, or pentylenetetrazol (PTZ)), electrical stimulation (e.g., perforant pathway, hippocampus, amygdala), models with genetic mutations (e.g., Dravet syndrome), and injury models (e.g., TBI, hyperthermia). To date, however, investigation of the impact of purinergic signalling on seizures and epilepsy has largely been restricted to the use of chemoconvulsants (i.e., KA and pilocarpine) in rodent models. For a more detailed description of in vivo models, please refer to [103,104] or other reviews written on this topic.

As mentioned in the previous section, among P2 receptors, the P2X7R has gained the most traction as possible drug target for epilepsy. Here, we will first provide a short summary of the evidence demonstrating the release of ATP, the main endogenous agonist of the P2X7R, during seizures and epilepsy, and on the role of P2X7R during seizures. This will be followed by a detailed discussion on the potential of P2X7R-targeting drugs in suppressing epilepsy-associated comorbidities. For a more extended description of the role of the P2X7R during epilepsy and the anticonvulsive and antiepileptic potential of drugs targeting this receptor, please refer to [19,25,26,90,105].

One of the first studies suggesting an association between increased extracellular ATP levels and increased neuronal activity was conducted by Heinrich et al., reporting increased extracellular ATP using depolarising high K^+^ concentrations in slices of rat hippocampus [106]. Increased extracellular ATP was, however, not observed when stimulating the Schaffer collateral [107]. More recently, using a rat model of intraperitoneal (i.p.) pilocarpine, Dona et al. provided the first in vivo evidence of increased extracellular ATP release during seizures and epilepsy [108]. In their study, the authors found that, while the purines ADP, adenosine monophosphate (AMP), and the anticonvulsant adenosine increased post-status epilepticus, no increase was found for ATP. In the same study, the authors also analysed purine levels in epilepsy. This analysis revealed that while ADP, AMP, and adenosine were decreased during the seizure-free period, the same purines increased following an epileptic seizure. This increase also included ATP. Progress has also been made with regard to the mechanisms of how ATP is released during seizures. Using rat hippocampal slices treated with the glutamate agonist ((S)-3,5-Dihydroxyphenylglycine), Lopartar et al. suggested that increases in extracellular ATP are mediated via the pannexin-1 channel [109]. Meanwhile, Dossie et al. showed, using resected tissue from patients with epilepsy, that extracellular ATP increased 80% during high-K^+^-induced ictal discharges and was suppressed by blocking pannexin-1. Of note, the same authors showed potent anticonvulsive effects when blocking pannexin-1 during KA-induced seizures in mice, suggesting drugs targeting ATP release mechanisms as novel treatment strategies for seizures and epilepsy [110]. Evidence suggesting a proconvulsant function of extracellularly released ATP came from studies showing that the injection of ATP, the ATP analogue 2,3-O-(4-benzoylbenzoyl)ATP (BzATP) or ADP into the lateral ventricle of mice, caused polyspiking on the electroencephalogram (EEG) (ATP) [111] and increased seizure severity during status epilepticus (BzATP and ADP) [112,113].

P2X7R expression has been shown to be increased across several models of status epilepticus. This includes following i.p. KA (hippocampus) [114] and intra-amygdala (i.a.) KA (hippocampus, cortex, striatum, thalamus, and cerebellum) [112,115,116] in mice and i.p. pilocarpine in rats (hippocampus) [117,118]. However, in contrast to the well-established increase in P2X7R expression post-status epilepticus, what cell types express the P2X7R remains to be fully established, in line with the wider ongoing controversy regarding its cell type-specific expression and function in the CNS as mentioned before [100,101]. While a neuronal expression of P2X7Rs post-status epilepticus has been suggested by two independent studies [112,115,117], others have failed to detect P2X7R in neurons showing P2X7R expression to be mainly restricted to microglia and oligodendrocytes [116,118,119]. P2X7R expression has also been shown to be increased during epilepsy in both preclinical rodent models and in TLE patients [115,117,120,121,122]. Similar to status epilepticus, while some studies have observed a neuronal expression in addition to the well-established expression on microglia [115,117,120,122], others failed to detect a neuronal expression of P2X7R during epilepsy [116].

Regarding its impact on seizures and epilepsy, several studies have reported anticonvulsant potential of P2X7R antagonists. This includes two studies using the i.a. KA-induced status epilepticus mouse model [112,115] and a more recent study using a rat model of coriaria lactone-induced status epilepticus [123]. Demonstrating that the anticonvulsant action is independent on the developmental stage, P2X7R antagonism also reduced seizures in two different animal models of infant seizures. This included a model where status epilepticus was induced via i.a. KA in 10-day old rats and a model of neonatal seizures where 7-day old mouse pups were subjected to 15 min of hypoxia [124,125]. Further in line with a proconvulsant function of P2X7Rs, Beamer et al. showed in a recent study that increased P2X7R expression leads to drug-refractoriness during i.a. KA-induced status epilepticus [126]. In other seizure models, including lithium-pilocarpine-induced status epilepticus, the maximal electroshock seizure threshold test and the PTZ seizure threshold test in mice failed, however, to replicate these anticonvulsant effects [118,127]. P2X7R antagonism, however, potentiated the anticonvulsant effects of carbamazepine in the electroshock seizure threshold and the PTZ seizure threshold test, suggesting P2X7R antagonists as potential adjunctive therapy [127]. P2X7R antagonism had no effect in a model of genetic absence epilepsy [128] and provided only a weak anticonvulsant effect in the intravenous PTZ, maximal electroshock and 6 Hz psychomotor seizure threshold tests [129]. In contrast to an anticonvulsant effect via blocking the P2X7R, Kim et al. showed that P2X7R-knockout mice had a more severe seizure phenotype in response to i.p. pilocarpine [130]. Interestingly, the same group observed that P2X7R deficiency did not alter seizure severity in the systemic KA mouse model or the picrotoxin mouse model [130].

In summary, while the P2X7R seems to contribute to seizures in models of status epilepticus, these effects seem to be minor or absent during acute non-damaging seizures. This may be explained by different levels of extracellular ATP concentrations reached under each condition. The P2X7R requires high amounts of ATP to be activated [131], which may be absent or insufficient during non-damaging seizures. Why seizures during status epilepticus are exacerbated via P2X7R antagonism in the pilocarpine model remains elusive. Possible explanations include differences in drug regimen and treatment windows; the most likely one is, however, the activation of different pathological pathways in the brain according to model used. Both models, KA and pilocarpine, are characterized by the development of status epilepticus, shortly following treatment with the proconvulsant, which is accompanied by neurodegeneration and activation of inflammation. The pilocarpine model, in contrast to the KA model, has, however, been associated with peripheral immune responses prior to the induction of status epilepticus and neuronal injury that most likely reflects a mixture of an ischemic and excitotoxic insult [103,132]. This may lead to differences in the degree of the activation/recruitment of different cellular/neuronal populations during status epilepticus, resulting in possible differences in responses according to where P2X7Rs are expressed. The P2X7R has, for example, been shown to depress mossy fiber-CA3 synaptic transmission [133]. Whether cell-type-specific effects of P2X7R activation lead to different responses during seizures requires, however, most likely the use of cell type-specific models (e.g., Cre-LoxP).

Suggesting an effect on the process of epileptogenesis, two studies showed P2X7R antagonism to decrease the mean kindling score in the rat PTZ kindling model [127,134]. Similar results were found using the rat pilocarpine model, where silencing of the P2X7R via siRNA resulted in reduced mortality, increased time to first spontaneous seizure, and reduced the number of spontaneous seizures [135]. In contrast, no effects on seizure-frequency via P2X7R antagonism was observed by Hong et al. using a rat model where epilepsy development was induced via lithium-pilocarpine [118]. In contrast, Rozmer et al. observed the development of a more severe epileptic phenotype following treatment with a P2X7R antagonists using the i.p.pilocarpine-induced status epilepticus model in mice [136], in line with the observed proconvulsive effects of P2X7R antagonism during status epilepticus when using the pilocarpine model.

To date, only two studies have analysed the impact of P2X7R antagonism on established epilepsy. In one study, using a model of multiple low doses of i.p. KA in rats, P2X7R antagonism reduced the severity of seizures without altering the seizure frequency [137]. In a second study, using the i.a. KA mouse model, P2X7R antagonism reduced the number of seizures per day, and this effect, remarkably, persisted following drug withdrawal, suggesting disease-modifying potential [122].

The mechanism behind how the P2X7R contributes to seizures and epilepsy remains to be established. The P2X7R has key roles in many physiological processes pertinent to seizure generation as well as epileptogenesis, including synaptic reorganization, the regulation of the BBB, cellular survival, circadian rhythms, neurogenesis, and inflammation [14,16,138,139,140]. Among these, neuroinflammation has attracted the most attention in the field [26]. P2X7R knockout or antagonism leads to lower levels of the proconvulsive cytokine IL-1β in the hippocampus following i.a. KA-induced status epilepticus [112,126] and reduces microgliosis and astrogliosis during epilepsy [122]. In line with the effects of P2X7R being mediated via its effects on microglia, we recently showed that P2X7R expression was increased in microglia during status epilepticus, at a time point when responses to anticonvulsants are reduced, and that P2X7R overexpression led to a pro-inflammatory phenotype in microglia during status epilepticus [112,126]. Moreover, the same study showed that pre-treatment of mice with Lipopolysaccharide increased P2X7R levels in the brain and reduced responsiveness to anticonvulsants during status epilepticus, which was overcome by either a genetic deletion of the P2X7R or the administration of P2X7R antagonists [126]. Suggesting a role on astrocytes, P2X7R antagonism protected from astrocyte cell death following i.p. pilocarpine-induced status epilepticus [141], and mice deficient in P2X7R presented decreased astroglia autophagy following i.p. KA [142]. It is, however, important to keep in mind that, whilst P2X7R-dependent activation of inflammation seems to be the most likely mechanism, the P2X7R contributes to numerous pathological processes, and more research is required to establish how this receptor causes seizures, critical for the design of future P2X7R-based therapies.

In summary, while there are still several gaps in our understanding of how P2X7R contributes to increased hyperexcitability states in the brain, compelling evidence has demonstrated its involvement in the generation of seizures and development of epilepsy, suggesting P2X7R-targeting drugs as novel anti-epileptic treatments.

## 5. Therapeutic Potential of Targeting P2X7R in Brain Diseases Commonly Associated with Epilepsy

As mentioned above, P2X7R activation has been shown to contribute to numerous diseases of the brain [22,24,143]. Consequently, P2X7R antagonism has been suggested as a potential novel treatment strategy for an array of different CNS conditions, including the most common comorbidities associated with epilepsy, as briefly discussed within this section.

### 5.1. P2X7R in Psychiatric Conditions

Depression, characterized by a sustained, pathological shift in mood and emotionality and accompanied by other symptoms affecting motivation, sleep, and appetite, is estimated to affect 5% of adults globally. It has a higher incidence in women and is a major precursor to suicide. Current antidepressant treatment includes selective serotonin and mixed serotonin–norepinephrine reuptake inhibitors and tricyclic anti-depressants; ~30% of patients remain, however, refractory to current treatment [144]. Evidence supporting a role of the P2X7R during depressive disorders stems from both human data and research carried out in animal disease models [28,145,146]. In humans, genetic population studies have suggested a possible association of the non-synonymous coding SNP rs2230912 in the *P2rx7* gene (Gln460Arg) with major depression disorder [147]. However, while some studies have reported a similar correlation [148,149], others have failed to confirm these results [150,151]. Data reported from preclinical studies, in contrast, have been more homogenous and conclusive. Here, mice expressing the human P2X7R carrying the Gln460Arg polymorphisms presented increased vulnerability to chronic social defeat stress and impaired sleep [152], suggestive of early stages of depression. More evidence suggesting a role of P2X7R during depression stems from studies showing effects of antidepressants on P2X7R function and expression [153,154,155]. Demonstrating its therapeutic potential, in vivo studies using mice knockout for P2X7R and P2X7R antagonists have shown anti-depressant-like effects in several models of depression (e.g., tail suspension test, forced swim test, chronic stress exposure) [156,157,158,159]. P2X7R antagonists were also effective in reverting depression-like behaviour in Flinders Sensitive Line (FSL) rats, a genetic model of depression [160].

The P2X7R has also been involved in bipolar disorder. Bipolar disorder is a chronic and severe mental disorder that presents in a cyclic course and has a prevalence of about 2.4% in the general population. Patients suffering from bipolar disorder type I usually present with episodes of severe mood swings from mania to major depression, while patients with bipolar disorder type II present milder forms of mood elevations, including milder episodes of hypomania that alternate with periods of severe depression. Pathological changes associated with bipolar disorder affect monoaminergic neurotransmission, including the dopaminergic, serotonergic and noradrenergic systems, redox imbalance, and neuroinflammation, with the latter being a matter of debate [161,162]. Evidence suggesting a role for the P2X7R during bipolar disorders stems from data showing P2X7R downregulation in a mouse model presenting a mania-like phenotype [163] and that P2X7R deletion or pharmacological targeting reverts increased locomotor activity induced by amphetamine treatment in rodents [164,165]. Genetic analysis carried out to identify a possible association between P2X7R polymorphism and bipolar disorder found that, similar to what was found for depression, while some studies reported a possible association, this could not be replicated by others [166,167,168].

Anxiety disorders are among the most prevalent and disabling psychiatric conditions. At least 25% of the population will suffer at least one episode of anxiety during their lifetime. These can include generalized anxiety disorder, panic disorders, specific phobias, separation anxiety disorder, social anxiety disorder, or agoraphobia, among others, with symptoms including anxiety, fear, and other mood disturbances. Treatment, mainly based on serotonin and norepinephrine reuptake inhibitors, benzodiazepines, and antidepressant drugs, remains, however, only partly effective [169]. Several studies have analysed the impact of P2X7R antagonism in the setting of anxiety; results have, however, been mixed, with pro- and anti-anxiolytic effects reported [156,170,171]. While genetic population studies have found some promising correlations between specific P2X7R polymorphisms and anxiety, these require confirmation in larger patient cohorts [148,149,172].

Schizophrenia is the most common psychiatric disorder requiring repeated periods of hospitalization over the patient’s life span. Prevalence of schizophrenia in the overall population is 1–2% with higher incidence reported for men. Patients may suffer from positive (e.g., hallucinations, delusions), negative (e.g., social withdrawal, lack of motivation), and cognitive symptoms. Current treatments may cause unwanted side effects and do not target all symptoms equally [173]. Consequently, the majority of patients remain permanently disabled, increasing the burden to the family and healthcare system. Similar to anti-depressants, several drugs used to treat schizophrenia (e.g., prochlorperazine, trifluoperazine) have been shown to alter P2X7R function [174]. While no association between schizophrenia and several P2X7R polymorphisms has been reported [175], recent studies in a mouse model using the chemical phencyclidine (PCP) have provided encouraging data showing that P2X7R knockout and treatment with specific P2X7R antagonists alleviated several schizophrenic-like parameters at the molecular and behavioural levels [27,176,177].

### 5.2. P2X7Rs and Other Common Co-Morbidities Associated with Epilepsy

In addition to psychiatric disorders, the P2X7R has been involved in several other diseases that show a higher incidence in patients with epilepsy when compared to the general population, including both CNS and non-CNS-related pathologies. A role of the P2X7R in AD is well-established [29,178]. AD is the most common form of dementia with over 50 million people suffering worldwide. AD is characterized by intracellular neurofibrillary tangles, made up of the hyperphosphorylated microtubule-associated protein tau, extracellular senile plaques, consisting mainly of amyloid-β protein deposits, widespread neurodegeneration, and synaptic loss and gliosis. Of note, neurofibrillary tangles seem to correlate better with the presence and severity of dementia than do plaques [179]. AD leads to the progressive loss of cognitive and behavioural abilities, eventually resulting in death. Presently, the main treatment options for AD are acetylcholinesterase inhibitors and NMDA-type glutamate receptor antagonists; both are, however, only symptomatic and minimally effective [180]. The P2X7R has been involved in α-cleavage of the amyloid precursor protein (APP), thereby altering the amount of toxic Aβ peptides [181,182]. In vivo studies have shown that P2X7R antagonism reduces the amyloid load in the brain, rescues cognitive deficits, and improves synaptic plasticity [183,184], demonstrating its therapeutic potential. More recent data also show that P2X7R antagonism protects against tau-induced pathology, both at a molecular and behavioural level [185,186,187]. While the mechanisms explaining how P2X7R activation leads to an AD-like phenotype are still to be determined, several pathways have been proposed to be regulated via P2X7R during AD and tauopathies, including phosphorylation of the tau kinase glycogen synthase 3β (GSK-3β), leading thereby to the hyperphosphorylation of tau [183,186] and inflammatory pathways, including a pathogenic CD8^+^ T cell recruitment [184]. Further in line with P2X7R contributing to tauopathies via driving inflammation, P2X7Rs were found to be highly expressed on microglia in mice overexpressing tau and in patients with tauopathies and P2X7R deficiency, reduced tau-related neuroinflammation, and microglia activation [186,187]. Nevertheless, regardless of how the P2X7R contributes to AD, these results strongly suggest that P2X7R is involved in memory loss and, consequently, identify the P2X7R as a possible therapeutic target to counteract cognitive decline seen, for example, in epilepsy [188]. Migraine is another common CNS disease associated with epilepsy, where P2X7R activation has been suggested to contribute to the pathology of the disease [189]. Cardiovascular conditions are among non-CNS diseases associated with epilepsy where P2X7R activation has also been suggested as a contributor. P2X7R has been shown to be expressed in cells of the heart, and its activation has been shown to contribute to several cardiovascular conditions, including hypertension, atherosclerosis, ischemia/reperfusion injury, and heart failure [190,191]. Increasing evidence now also suggests a role of the P2X7R in diabetes [192], gastrointestinal diseases [193], and bone diseases [194], among others. These peripheral comorbidities are most likely due to P2X7R localization on immune cells such as lymphocytes and monocytes/macrophages or, in the case of osteoporosis, on macrophage-like osteoklasts [195].

Taken together, mounting evidence suggests P2X7R-based treatment as a therapeutic avenue for numerous pathological conditions, which, in turn, implies P2X7R signalling as a shared underlying pathology. What pathological pathways are activated via P2X7Rs in each disease remains to be established. While inflammation represents the most obvious one, with the dominant expression of the P2X7R within the immune system (including microglia) [88], it is important to bear in mind that the P2X7R has been shown to be involved in several other physiological and pathological pathways. This includes the regulation of synaptic plasticity, neurogenesis, and changes in neurotransmitter release, including the release/uptake of serotonin (5-HT), noradrenaline (NA), glutamate, and GABA [146], thereby possibly contributing to the altered brain neurochemistry observed under stressful conditions. It is, however, important to remember that inflammation has been shown to affect all of these processes [196]. Pinpointing, therefore, the exact mechanisms of action of the P2X7R during diseases will most likely require more sophisticated methods than the use of P2X7R antagonists and global P2X7R knock-out models (e.g., the use of cell type-specific knockouts). Another intriguing unanswered question is whether P2X7R-driven pathologies are the cause or consequence of brain diseases. While in most cases contributions of P2X7Rs to disease progression are probably secondary and result most likely as a consequence of an increased inflammatory tissue tone, an increased P2X7R activation may, however, lead to the development of other associated comorbidities. Finally, epilepsy, besides presenting with increased inflammation in the brain, is also characterized by increased systemic inflammation [197], possibly leading to comorbidities that involve peripheral organs such as the heart or the gastrointestinal system. Of note, we have recently shown the P2X7R to be increased in the blood of epilepsy patients [198], potentially contributing to systemic inflammation and its deleterious consequences on peripheral organs. In the final section of this review, we discuss in detail studies investigating the impact of targeting the P2X7R on epilepsy-associated comorbidities.

## 6. P2X7R as Treatment Target for Epilepsy-Associated Comorbidities

While the majority of studies investigating the therapeutic potential of targeting the P2X7R in the setting of epilepsy have been restricted to its impact on seizures and the development of epilepsy, several studies have reported beneficial effects beyond seizure suppression (Table 1).

One of the first studies analysing the impact of P2X7R antagonism on epilepsy-associated comorbidities was carried out by Soni et al. in 2015 using a model of PTZ-induced kindling in rats [134]. Rats were treated via a systemic administration of PTZ (30 mg/kg) each consecutive day for 27 days. Rats subjected to PTZ kindling showed an increased seizure susceptibility and reduced motor co-ordination and cognitive skills as analysed via the Morris Water Maze, an object-recognition task, and Rotarod. To test whether P2X7R antagonism protects against seizures and motor and cognitive deficits, rats were treated with the non-specific P2X7R antagonist Brilliant blue G (BBG) (15 and 30 mg/kg) once daily for 27 days starting at the same time as treatment with PTZ. Behavioural tests were performed one week following drug withdrawal. Suggesting protective properties of P2X7R antagonism, rats treated with BBG presented a significant reduction in the mean kindling score and restored behavioural performance, including cognition and motor function. Interestingly, the authors further showed that a combination treatment with BBG and the glutamate transporter-1 up-regulator ceftriaxone enhanced the protective effects observed with BBG or ceftriaxone alone.

In the same year, Jimenez-Mateos et al. published a study providing further evidence, even though indirect, of an involvement of P2X7R in the development of epilepsy and associated comorbidities [201]. In this study, the authors identified the microRNA-22 (miR-22) to negatively regulate P2X7R expression following i.a. KA-induced status epilepticus and during epilepsy. Here, mice were treated with a specific miR-22 inhibitor (antagomir-22), which resulted in an increased P2X7R expression in the brain, including the hippocampus. While antagomir-22 treated mice experienced similar seizure severity during status epilepticus, these mice developed a more sever epileptic phenotype when compared to scramble-treated epileptic control mice. To test any impacts on behaviour, mice were also analysed using the novel object relocation test. This revealed that epileptic mice treated with antagomir-22 spent less time in open areas during monitoring when compared to scramble-treated epileptic mice and non-epileptic control animals, suggesting increased anxiety. Epileptic mice treated with antagomir-22 also failed to show a preference for any object in the novel object re-location test, suggesting cognitive deficits.

First evidence suggesting a contribution of the P2X7R to depression during epilepsy stems from a study by Hong et al. in 2020 [118]. In their study, the authors used a rat model, where status epilepticus was induced via an i.p. pilocarpine injection shortly following an administration of lithium. Rats were treated with the P2X7R antagonists BBG (100 mg/kg) once daily starting before the induction of status epilepticus for the duration of the entire experiment. While epileptic rats showed an increased depressive-like behaviour and anxiety measured via the open field test, sucrose preference test, forced swimming test, and elevated-plus maze, mice treated with BBG showed a similar behaviour compared with non-seizure control rats, suggesting both anti-depressive and anti-anxiolytic effects via P2X7R antagonism. Interestingly, treatment with P2X7R antagonists had no apparent effect on epilepsy development with both vehicle- and BBG-treated rats showing similar frequency and severity of epileptic seizures. This suggests that, while P2X7R antagonism my not always lead to a reduction/suppression in seizures, it may still be beneficial for the treatment of other associated conditions, further supporting the idea of P2X7R antagonism as adjunctive therapy.

Finally, Jamali-Raeufy et al. provided evidence of a beneficial effect of P2X7R antagonism on cognition during epilepsy [202]. Here, the authors used the intrahippocampal (i.h.) KA model in rats and, as in the previous study, used BBG (2 nM, intracranial, administered once 30 min prior to the induction of status epilepticus) to block P2X7Rs. When analysed 2 weeks post-status epilepticus, rats treated with BBG performed significantly better on the Y-maze task compared to epileptic vehicle-treated rats. Of note, in their study, BBG also reduced the severity of status epilepticus, which may be the reason for a better performance of BBG-treated rats during memory tests once epileptic.

In summary, while these data support a role of the P2X7R in the development of epilepsy-associated comorbidities, there are several key questions which will have to be addressed before definite conclusions can be drawn. To date, to our knowledge, only four studies using animal models of epilepsy have analysed a possible role of the P2X7R in epilepsy-related comorbidities. While seizures are the main focus when investigating the therapeutic potential for any novel ASD, epilepsy, rather than being purely a seizure disorder, represents a spectrum of different pathological conditions. Moreover, some drugs may fail to tackle seizures but may still have an effect on other associated pathologies. Thus, future studies should be designed to evaluate the complete therapeutic picture of new treatments including their effectiveness in suppressing seizures and associated comorbidities. A more comprehensive analysis in terms of comorbidities would, however, require that epilepsy models are thoroughly characterized in terms of not only seizure occurrence but also associated comorbidities. A more difficult question to answer would be whether effects on comorbidities are due to a reduction in seizures or whether this is a direct effect on comorbidities. Interestingly, the study carried out by Hong et al. showed improved cognition in P2X7R antagonist-treated rats without any noticeable effect on seizures [118]. This suggests that P2X7R antagonist-mediated improvements in cognition are independent on seizure modulation. To date, most studies analysing the role of P2X7R in comorbidities used the non-specific P2X7R blocker BBG. BBG, although a potent antagonist at the P2X7R with an IC_50_ value of approximately 400 nM, also antagonizes several other receptors/channels such as P2X receptors (e.g., P2X1, P2X2, P2X3, and P2X4) and voltage-gated sodium channels, although at much higher concentrations [203]. In addition, the usual therapeutic regime requires multiple applications of BBG instead of a single application. Therefore, results should be confirmed using more specific and brain penetrant P2X7R antagonists, which are now available [122,127,204]. Furthermore, the analysis of the role of P2X7Rs in epilepsy-associated comorbidities focused on CNS-related comorbidities. Epilepsy is, however, associated with several non-CNS comorbidities (e.g., cardiovascular conditions), where P2X7R antagonism has been proposed as possible treatment (see Section 5). Are there P2X7R polymorphisms which may favour both seizures and comorbidities? Several genetic studies have suggested associations between P2X7R polymorphisms and psychiatric conditions. Do some of these polymorphisms also predispose to suffer from epilepsy? For example, the P2X7R rs208294 His155Tyr polymorphism has been associated with childhood febrile seizure susceptibility [205] and several psychiatric conditions, including anxiety [206] and bipolar disorder [149].

## 7. Conclusions

We now have a substantial body of evidence demonstrating the therapeutic potential of targeting the purinergic P2X7R during epilepsy. This includes anti-convulsive and anti-epileptogenic effects provided by P2X7R antagonists and the disease-modifying effects observed when given during established epilepsy. Critically, P2X7R antagonism has also been shown to be effective in treating the most common comorbidities associated with epilepsy, thus representing a novel therapeutic avenue reducing seizure burden while also tackling quality-of-life-reducing comorbidities (Figure 3). Future studies should, however, be designed to establish whether the co-morbidities improve because epilepsy is cured or whether this is an independent effect on the co-morbidities. Nevertheless, regardless of how P2X7R-based treatments affect comorbidities, with drugs targeting the P2X7R quickly moving from pre-clinical to clinical trials stage for CNS diseases [94,207], the first clinical trials targeting this receptor in epilepsy should be expected in the near future.

## Figures and Tables

**Figure 1 ijms-23-02380-f001:**
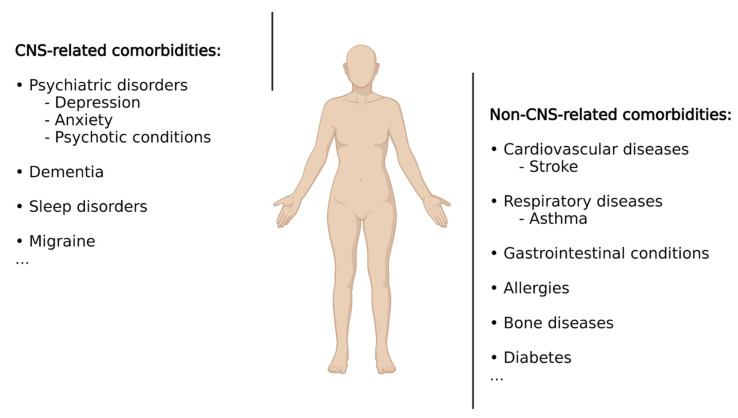
Epilepsy-associated co-morbidities. Schematic representation showing the most common comorbidities associated with epilepsy including CNS- and non-CNS-related conditions. Created with BioRender.com (accessed on 17 February 2022).

**Figure 2 ijms-23-02380-f002:**
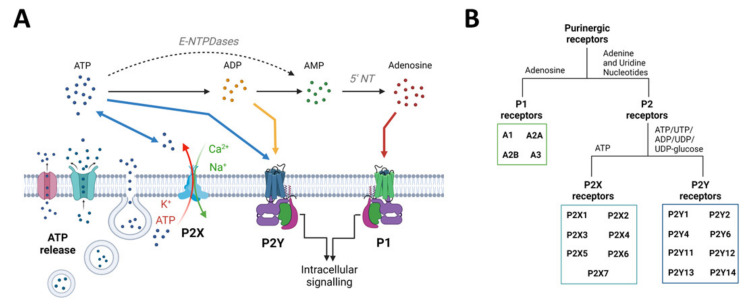
Purinergic signalling overview: from ATP release mechanisms to ATP receptors. (**A**) ATP can be released from neurons and glia via transporters, membrane channels, and exocytosis. ATP can also be released via P2X7 channels (P2X7R). Once released, ATP is converted into adenosine through the intermediates ADP and AMP via ectoenzymes such as NTPDases, NPPases, and alkaline phosphatase. Extracellular ATP can act on P2X (ligand gated) and P2Y (G protein coupled) receptors. ADP, the first breakdown product, can also act via certain subtypes of P2Y receptors. Adenosine acts via P1 (G protein coupled) receptors. (**B**) Schematic overview of purinergic receptors is divided into adenosine-sensitive P1 receptors, which are further subdivided into A_1_, A_2A_, A_2B_, and A_3_ receptors, and nucleotide-sensitive P2 receptors, which are divided into inotropic P2X and metabotropic P2Y receptors. There are seven mammalian P2X receptor subtypes, which all respond to ATP. The eight to date identified P2Y receptor subtypes respond to ATP, ADP, UTP, UDP, or UDP-glucose, depending on the subtype. Created with BioRender.com (accessed on 17 February 2022).

**Figure 3 ijms-23-02380-f003:**
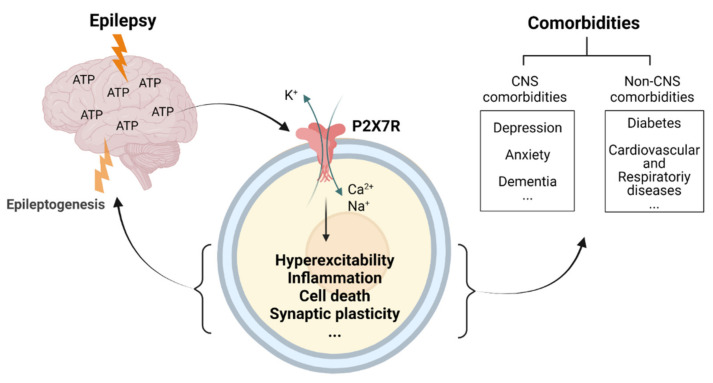
The ATP-P2X7 receptor axis as major convergence pathway linking hyperexcitability and associated comorbidities in epilepsy. Seizures and epilepsy lead to high extracellular concentrations of ATP in the brain-activating P2X7Rs. P2X7R activation in turn promotes several pathological mechanisms such as inflammation, cell death, aberrant synaptic plasticity, and changes in the release and uptake of neurotransmitters, driving epileptogenesis and contributing to the development of several comorbidities. Created with BioRender.com (accessed on 17 February 2022).

**Table 1 ijms-23-02380-t001:** Selected examples of in vivo studies investigating the impact of P2X7R signalling on seizures and epilepsy-related co-morbidities.

Model/Disease Stage	Strategy of Targeting P2X7R	Impact on Seizures/Epilepsy	Impact on Brain Pathology	Impact on Comorbidities/Tests Used	Reference
*Status epilepticus* KA (25 mg/kg, i.p.), picrotoxin (5 mg/kg, i.p.), pilocarpine (150, 175, 200, 225, or 250 mg/kg, i.p.) in mice (males)	P2X7R KO mice P2X7R antagonist(s): OxATP (5 mM), A-438079 (10 μM) and A740003 (10 μM) delivered over 1 week via an osmotic mini-pump (model 1007D) before seizure induction	Increased seizure susceptibility post-pilocarpine via P2X7R KO and P2X7R antagonism. No effects on seizures in the KA and picrotoxin model.	Not studied.	Not studied.	[130]
*Status epilepticus* Pilocarpine (380 mg/kg, i.p.) in rats (males)	P2X7R agonist(s): B_Z_ATP (5 mM) via an osmotic mini-pump (0.5 μL/h for 1 week) P2X7R antagonist(s): OxATP (5 mM), A-740003 (5 mM), and A-438079 (10 μM) delivered via an osmotic mini-pump (0.5 μL/h for 1 week)	Not studied.	P2X7R agonism: Reduced neurodegeneration; P2X7R antagonism: Increased neurodegeneration, reduced astroglial death and reduced infiltration of neutrophils.	Not studied.	[141,199,200]
*Status epilepticus* KA (0.3 µg, i.a.) in mice (males)	P2X7R KO mice P2X7R angonist(s): B_Z_ATP (0.1 nmol, i.c.v.); P2X7R antagonists: A438079 (1.75 nmol, i.c.v.), BBG (1 pmol, i.c.v.); P2X7R antibody (APR-008, 0.7 mg/mL, i.c.v.)	P2X7R agonism: Increased seizure severity P2X7R antagonism: Reduced seizure severity	P2X7R antagonism-mediated neuroprotection in hippocampus and cortex.	Not studied.	[112,115]
*Status epilepticus* Coriaria lactone (40 mg/kg, i.m.) in rats (males)	P2X7R angonist(s): B_Z_ATP (5 mM, i.c.v, 2 µL). P2X7R antagonist(s): BBG (1, 5, 10 µg; i.c.v.), A438079 (10 µM, ic.v., 2 µL), and A740003 (10 µM, i.c.v., 2 µL) (pre-treatment)	P2X7R antagonism-mediated reduction in seizure severity during status epilepticus.	P2X7R antagonism reduced inflammation, neuronal damage, astrogliosis, and microgliosis.	Improved cognitive function 2 weeks post-status epilepticus (Morris water maze test).	[123]
*Status epilepticus* KA (0.3 µg for C57 and 0.2 µg FVB background, i.a.) in mice (male and female)	P2X7R KO and P2X7R overexpressing mice P2X7R antagonist(s): AFC-5128 (50 mg/kg, i.p.) or ITH15004 (1.75 nmol, i.c.v.) at time of anticonvulsant treatment	No effect on status epilepticus. P2X7R overexpression caused unresponsiveness to ASDs; P2X7R KO and antagonism potentiated effects of ASDs.	No effects on cell death. Increased inflammation in P2X7R overexpressing mice post-status epilepticus.	Not studied.	[126]
*Acute seizures* (focal, generalized, and generalized tonic–clonic) Timed PTZ infusion test (1% PTZ 2 mL/min, i.v.); MES-T; 6 Hz electroshock-induced seizures (0.2 ms square pulse at 6 Hz for 3 s) in mice (males)	P2X7R antagonist(s): BBG (acute, 100–400 mg/kg, i.p., 30 min prior to test), (sub-chronic, 25–100 mg/kg, i.p., once daily for seven consecutive days)	Reduced seizures during 6 Hz test (focal seizure). No significant anticonvulsive effects of BBG in i.v. PTZ and MES-T test (generalized and generalized tonic-clonic seizures).	Not studied.	Not studied.	[129]
*Acute seizures* (absence seizures) WAG/Rij rats (inbred strain of rats with genetic absence epilepsy) (males)	P2X7R agonist(s): i.c.v. B_Z_ATP (50 μg and 100 μg) P2X7R antagonist(s): i.c.v. A-438079 (20 μg and 40 μg)	No effects of P2X7R agonists or antagonists on spike-wave discharges (SWDs).	Not studied.	Not studied.	[128]
*Acute seizures* MES-T (inusoidal pulses 4–14 mA, 50 Hz, 0.2 s duration) and PTZ-T (87 mg/kg, s.c.) in mice (males) *Epileptogenesis* PTZ kindling (35 mg/kg, i.p.) in rats (males) for 25 days	P2X7R antagonist(s): JNJ-47965567 (15 or 30 mg/kg), AFC-5128 (25 or 50 mg/kg), BBG (50 mg/kg), transhinone (30 mg/kg), all drugs injected i.p.	No effects on acute seizures alone; Reduced seizure severity in combination with carbamazepine; AFC-5128 and JNJ-47965567 showed a significant and long-lasting delay in kindling development	Reduced glial activation.	Not studied.	[127]
*Epileptogenesis* PTZ kindling (30 mg/kg, i.p.) every second day for 27 days in rats (sex not specified)	P2X7R antagonist(s): BBG (15 and 30 mg/kg, i.p.) 30 min before PTZ injection	Reduced seizure score during kindling.	Increased glutathione levels and reduced lipid peroxidation and nitrite levels.	Improved motor performance (Rotarod) and cognitive deficits (Morris Water Maze, Object recognition task) 35–41 days after kindling start.	[134]
*Epileptogenesis* KA (3 µg, i.a.) in mice (males)	Injection of Antagomir-22 (0.5 nmol, i.c.v.) 1 day before induction of status epilepticus	Increased seizure frequency during epilepsy.	Increased P2X7R expression in Antagomir-22 treated mice accompanied by increased neuroinflammation.	Increased anxiety (Open field) and memory deficits (Object location task) 14 days post-status epilepticus in Antagomir-22 treated mice.	[201]
*Epileptogenesis* Pilocarpine (370 mg/kg, i.p.) in rats (males)	P2X7R antagonist(s): AZ10606120 (3 µg/2 µL, i.c.v.) post-SE/BBG (50 mg/kg, i.p.) 1 injection per day for 4 days poststatus epilepticus	P2X7R antagonisms increased seizure number and seizure severity during epilepsy.	Neuroprotection mediated via P2X7R antagonism post-status epilepticus.	Not studied.	[136]
*Epileptogenesis* Pilocarpine (370 mg/kg, i.p) in rats (males)	P2X7R-targeting siRNA (1 μg of siRNA per animal, i.c.v.) 6 h after onset of status epilepticus	Delayed seizure onset and reduced seizure frequency during epilepsy.	P2X7R antagonisms mediated neuroprotection in hippocampus, reduced edema, reduced mortality following status epilepticus.	Not studied.	[135]
*Epileptogenesis* KA (4 µg, i.h.) in rats (males)	P2X7R antagonist(s): BBG (2 nM, i.c.v) 30 min prior to induction of status epilepticus.	Reduced seizure severity following pre-treatment with BBG.	Neuroprotection, reduced aberrant mossy fiber sprouting and neuroinflammation.	Improved spatial memory (Y-maze)	[202]
*Epilepsy* Lithium chloride (127 mg/kg, i.p.) followed by pilocarpine (30 mg/kg, i.p., followed by 10 mg/kg, i.p. every 30 min until development of convulsive seizures) in rats (males)	P2X7R antagonist(s): BBG (100 mg/kg, i.p.) 24 h before status epilepticus and once daily post-status epilepticus	No effect on status epilepticus and epilepsy development.	Reduced microglia activation and neuronal loss.	Anti-depressive (Sucrose preference test, Forced swimming test) and anti-anxiety (Open field, Elevated plus maze) effects via P2X7R antagonism.	[118]
*Epilepsy* Multiple low-dose KA (total KA = 22.2 ± 2.02 mg/kg, i.p.) in rats (males)	P2X7R antagonist(s): JNJ-47965567 during 1 week via osmotic mini-pump (0.6 g/kg/2mL)	Decreased seizure severity without changes in total number of seizures.	P2X7R antagonism had no impact on inflammation.	Not studied.	[137]
*Epilepsy* KA (3 µg, i.a.) in mice (males)	P2X7R antagonist(s): JNJ-47965567 (30 mg/kg, i.p.) twice daily for 5 days during epilepsy	Reduced seizure frequency during treatment following drug withdrawal.	Decreased inflammation (astrogliosis and microgliosis).	Not studied.	[122]

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
