# Peer review of "Beyond Seizure Control: Treating Comorbidities in Epilepsy via Targeting of the P2X7 Receptor"

_ijms, 2022, doi:10.3390/ijms23042380_

Round 1

Reviewer 1 Report

This is a very well written manuscript on the evidence suggesting drugs targeting the P2X7R as novel treatment strategy for epilepsy with a particular focus of its potential impact on epilepsy-associated comorbidities. I have only few suggestions of minor importance to improve the manuscript.

  1. Figure 2: P2X receptors - please correct P2X51.
  2. Please correct the spelling of Löscher through the manuscript.
  3. A recent paper 10.1016/j.pharmthera.2021.107821 could be cited as it contains information relevant to this manuscript.

Author Response

Thank you very much for the positive feedback. We have included the changes suggested in our revised manuscript as outlined below.

  1. Figure 2: P2X receptors - please correct P2X51.

Thank you for spotting this mistake. This has been corrected.

  1. Please correct the spelling of Löscher through the manuscript.

This has been corrected accordingly.

  1. A recent paper 10.1016/j.pharmthera.2021.107821 could be cited as it contains information relevant to this manuscript.

This reference has been included in our revised manuscript (Section 5.1).

“Evidence supporting a role of the P2X7R during depressive disorders stems from both human data and research carried out in animal disease models [28, 145, 146].”

Reviewer 2 Report

This manuscript clearly and accurately reviews the role of the ATP-gated P2X7 receptor channel in epilepsy and comorbidities. To the best of my knowledge this has not been described in detail by any author, except by the authors, in part, in neonatal epilepsy (see DOI: 10.3390/ijms21217832). I have only minor editorial concerns.

  1. Page 3, paragraph 1: Some of the font differs to the rest manuscript and should be corrected.
  2. Page 3, paragraph 2: The first sentence lacks clarity. Consider inserting "with epilepsy" after "50% of patients".
  3. Various terms are unnecessarily capitalized (p3, "Amyloid"; p3, Damage-Associated Molecular Patterns; p6 "Dihydrox-"; p10, "Glycogen Synthase"; p15, "Glutamate" and possibly "Elevated-plus") and should be corrected to sentence case. Please check for other examples also.
  4. Table 1 is cited on page 6, but does not appear immediately after this as per IJMS Guidelines. Either delete "Table 1" here or move Table 1.
  5. Page 6, paragraph 3: Delete space between "anticonvulsant    adenosine".
  6. Page 10, paragraph 3: Italicize "I" in "In vivo".
  7. Table 1: Molar concentrations rather than molar amounts are presented on occasion (e.g. 5 mM OxATP), which is less informative. If molar amounts are available in cited publications, please update table.
  8. Table 1 (final page): "pilocarpine" (twice), "lithium" are not capitaized correctly and should be corrected. Also is a symbol missing in "KA(4     g, i.h.)"?
  9. The Abbreviations (page 17) are incomplete, with many missing and should be updated. Additionally injection routes remain in full throughout text, when arguably they should be abbreviated as per Table 1. Please revise all abbreviations for consistency, including ensuring all abbreviations in Table 1 appear in Abbreviations (page 17).
  10. Volume/page/manuscript number details missing from Ref. 198.

Author Response

We would like to thank the reviewer for the positive comments. We have included all suggested changes by the reviewer as outlined below.

  1. Page 3, paragraph 1: Some of the font differs to the rest manuscript and should be corrected.

This has been corrected.

  1. Page 3, paragraph 2: The first sentence lacks clarity. Consider inserting "with epilepsy" after "50% of patients".

The sentence has been amended according to the reviewer’s suggestions:

“Psychiatric disorders are one of the most frequently reported comorbidities, estimated to occur in 25% - 50% of patients with epilepsy [4], and being particularly prevalent in patients with TLE and drug-refractory epilepsy [35-38].”

  1. Various terms are unnecessarily capitalized (p3, "Amyloid"; p3, Damage-Associated Molecular Patterns; p6 "Dihydrox-"; p10, "Glycogen Synthase"; p15, "Glutamate" and possibly "Elevated-plus") and should be corrected to sentence case. Please check for other examples also.

Thank you for pointing this out. This has been corrected throughout the manuscript.

  1. Table 1 is cited on page 6, but does not appear immediately after this as per IJMS Guidelines. Either delete "Table 1" here or move Table 1.

Thank you. We have deleted Table 1 from page 6, as we think the table is better placed towards the end of the manuscript.

  1. Page 6, paragraph 3: Delete space between "anticonvulsant    adenosine".

Space has been deleted.

  1. Page 10, paragraph 3: Italicize "I" in "In vivo".

This has been corrected.

  1. Table 1: Molar concentrations rather than molar amounts are presented on occasion (e.g. 5 mM OxATP), which is less informative. If molar amounts are available in cited publications, please update table.

The Table has been updated according to the reviewers suggestions. Where possible, we have included the quantity which was injected or included the model of pumps if no further information was given within the original manuscript.

  1. Table 1 (final page): "pilocarpine" (twice), "lithium" are not capitaized correctly and should be corrected. Also is a symbol missing in "KA(4     g, i.h.)"?

This has been corrected.

  1. The Abbreviations (page 17) are incomplete, with many missing and should be updated. Additionally injection routes remain in full throughout text, when arguably they should be abbreviated as per Table 1. Please revise all abbreviations for consistency, including ensuring all abbreviations in Table 1 appear in Abbreviations (page 17).

The abbreviation list has been updated accordingly. We have also included the same abbreviations in text and table.

Abbreviations: AD, Alzheimer’s disease; ADP, Adenosine diphosphate; AMP, Adenosine monophosphate; ASD, Anti-seizure drugs; ATP, Adenosine triphosphate; BBB, Blood-brain barrier; BBG, Brilliant blue G; BzATP, 2 ,3 -O-(4-benzoylbenzoyl)ATP; i.a., intra-amygdala; CNS, Central nervous system; EEG, Electroencephalogram; GABA, γ-aminobutyric acid; i.c.v., intracerebroventricular; i.h., intrahippocampal; IL-1β, Interleukin-1β; i.p., intraperitoneal; i.v., intravenous; KA, kainic acid; P2X7R, P2X7 receptor; PCP, Phencyclidine; PTZ, Pentylenetetrazol; s.c., subcutaneous; SNP, single nucleotide polymorphisms; TBI, Traumatic brain injury; TLE, Temporal lobe epilepsy; UDP, Uridine diphosphate; UTP, Uridine triphosphate

  1. Volume/page/manuscript number details missing from Ref. 198.

This has been corrected.